# PeerJ

# Decadal changes and delayed avian species losses due to deforestation in the northern Neotropics

David W. Shaw[1], Patricia Escalante[2], John H. Rappole[3],
Mario A. Ramos[5], Richard J. Oehlenschlager[4], Dwain W. Warner[5] and
Kevin Winker[1]

[1] University of Alaska Museum, Fairbanks, AK, USA
[2] Colección Nacional de Aves, Instituto de Biología, Universidad Nacional Autónoma de México,
Ciudad Universitaria, Copilco, Coyoacan, Mexico
[3] Smithsonian's National Zoological Park, Conservation and Research Center, Front Royal, VA,
USA
[4] Science Museum of Minnesota, St. Paul, MN, USA
[5] Deceased

## ABSTRACT

How avifauna respond to the long-term loss and fragmentation of tropical forests is a critical issue in biodiversity management. We use data from over 30 years to gain insights into such changes in the northernmost Neotropical rainforest in the Sierra de Los Tuxtlas of southern Veracruz, Mexico. This region has been extensively deforested over the past half-century. The Estación de Biología Tropical Los Tuxtlas, of the Universidad Nacional Autónoma de México (UNAM), protects a 640 ha tract of lowland forest. It became relatively isolated from other forested tracts between 1975 and 1985, but it retains a corridor of forest to more extensive forests at higher elevations on Volcán San Martín. Most deforestation in this area occurred during the 1970s and early 1980s. Forest birds were sampled on the station and surrounding areas using mist nets during eight non-breeding seasons from 1973 to 2004 (though in some seasons netting extended into the local breeding season for some species). Our data suggested extirpations or declines in 12 species of birds subject to capture in mist nets. Six of the eight species no longer present were captured in 1992–95, but not in 2003–2004. Presence/absence data from netting and observational data suggested that another four low-density species also disappeared since sampling began. This indicates a substantial time lag between the loss of habitat and the apparent extirpation of these species. Delayed species loss and the heterogeneous nature of the species affected will be important factors in tropical forest management and conservation.

## INTRODUCTION

Deforestation is one of the main threats to biodiversity conservation. Forest loss and fragmentation have caused declines or local extinctions among animal species at many locations (*Turner, 1996*; *Fahrig, 2003*; *Dirzo & Raven, 2003*). Local population declines and extirpations may be the most important leading indicators of biodiversity loss

Corresponding author
Kevin Winker,
kevin.winker@alaska.edu

(*Ceballos & Ehrlich, 2002*; *O'Grady et al., 2004*). Bird losses have been documented in many forest systems (e.g., *Willis, 1974*; *Willis, 1979*; *Leck, 1979*; *Karr, 1982*; *Bierregaard & Lovejoy, 1989*; *Kattan, Alvarez-López & Giraldo, 1994*; *Robinson, 1999*; *Sodhi, Liow & Bazzaz, 2004*; *Ferraz et al., 2007*; *Patten, Gómez de Silva & Smith-Patten, 2010*; *Laurance et al., 2011*). Perhaps nowhere has this phenomenon been more noticeable than among tropical forests, where species losses have been documented in numerous taxonomic groups (e.g., *Zimmerman & Bierregaard, 1986*; *Powell & Powell, 1987*; *Malcolm, 1988*; *Pahl, Winter & Heinsohn, 1988*; *Becker, Moure & Peralta, 1991*; *Daily & Ehrlich, 1995*; *Brook, Sodhi & Ng, 2003*; *Dirzo & Raven, 2003*; *Stuart et al., 2004*; *Robinson & Sherry, 2012*). Species losses can occur at the landscape or patch levels and depend on the intensity of the change in forest cover, the distance to and size of other forest fragments, shape and size of the fragment, and other factors (*Robbins, 1980*; *Lovejoy et al., 1984*; *Lovejoy et al., 1986*; *Rolstad, 1991*; *Andrén, 1994*; *Faaborg et al., 1995*; *Lees & Peres, 2006*; *Barlow et al., 2006*; *Patten & Smith-Patten, 2011*; *Robinson & Sherry, 2012*). Tropical forest species, which often occur in small, low-density populations, may be particularly vulnerable to extirpation (*Terborgh & Winter, 1980*; *Pimm, Jones & Diamond, 1988*; *Stotz et al., 1996*).

Relatively few studies have assessed changes through decades, however (*Ewers & Didham, 2006*). And although deforestation and fragmentation can occur over a short period, some time may pass before species begin to disappear from an affected area (*Leigh, 1975*; *Leigh, 1981*; *Karr, 1982*; *Tilman et al., 1994*; *Brooks, Pimm & Oyugi, 1999*). Thus, to fully document the impact of deforestation on a forest community, a site must be studied for a substantial period of time after habitat alteration has occurred. Detailing the process of local population decline and extirpation over time provides invaluable information about species' abilities to cope with habitat fragmentation. It also informs us about how community composition itself may be resistant to change, its degree of resilience following change, and how or if it stabilizes following this disturbance.

Studies of species losses in birds have used a variety of methods, including comparing species richness in different-sized fragments (*Willis, 1979*; *Nemark, 1991*; *Blake, 1991*), comparison of species composition at a site pre- and post-fragmentation (*Willis, 1974*; *Leck, 1979*; *Bierregaard & Lovejoy, 1989*; *Kattan, Alvarez-López & Giraldo, 1994*; *Patten & Smith-Patten, 2011*), and experimental fragmentation (*Lovejoy et al., 1986*; *Bierregaard & Lovejoy, 1988*; *Bierregaard & Lovejoy, 1989*; *Ferraz et al., 2003*; *Ferraz et al., 2007*; *Laurance et al., 2011*), and have often included scattered survey data prior to fragmentation (*Willis, 1974*; *Leck, 1979*; *Kattan, Alvarez-López & Giraldo, 1994*; *Robinson, 1999*; *Patten, Gómez de Silva & Smith-Patten, 2010*; *Patten & Smith-Patten, 2011*). Many of these studies have relied on qualitative visual and audio survey techniques, with multiple observers, though such techniques can allow cryptic and low-density species to be overlooked (*Whitman, Hagan & Brokaw, 1997*). Additionally, observer skills and intensity of sampling may vary among surveys.

Mist netting offers the most consistent and quantitative method available to sample birds among years (*Rappole, Winker & Powell, 1998*). However, mist nets have documented weaknesses; the most relevant is the limited stratum and size of birds they can effectively

sample (*Remsen & Good, 1996*; *Whitman, Hagan & Brokaw, 1997*; *Rappole, Winker & Powell, 1998*). This is particularly noticeable in structurally diverse habitats such as tropical rainforests, where probability of detection using mist nets is unknown for most species. Mist net studies in the Neotropics are therefore biased toward understory, small- to mid-sized passerines. While mist nets, unlike other methods, are less prone to observer bias and variability, we augmented our analyses of netting data that suggested species losses with presence-absence observational data (daily checklists in later years); this becomes particularly important for low-density species and for those not readily captured.

The Sierra de Los Tuxtlas of southern Veracruz, Mexico provides a textbook case of deforestation. This small range of volcanic mountains is home to the northernmost Neotropical rainforest (*Pennington & Sarukhan, 1968*; *Dirzo & Miranda, 1991*). The region has lost more than 90% of its forests in the past century, with the majority of that loss occurring in the lowlands over the past fifty years (*Dirzo & Garcia, 1992*; *Rappole, Powell & Sader, 1994*; *Winker, 1997*). Our study compares eight seasons of mist net sampling from Los Tuxtlas over the course of more than thirty years. This allows us to at least partly answer the question of how species composition and relative abundance changed in and around a conserved core of local rainforest habitat on a decadal scale.

## METHODS

The Sierra de Los Tuxtlas is located in southern Veracruz, Mexico, 90 km southeast of Veracruz city. This range of mountains lies in the northwestern portion of the Isthmus of Tehuantepec and is isolated from the Sierra Madre Oriental by extensive lowlands. Los Tuxtlas encompass approximately 4,200 km$^2$, and the range is dominated by Volcán San Martín and Volcán Santa Marta, each reaching more than 1,500 m elevation. The Gulf of Mexico lies a short distance from the mountains to the north and east. The northernmost Neotropical evergreen rainforest formerly dominated the habitat in the region (*Andrle, 1966*; *Pennington & Sarukhan, 1968*; *Dirzo & Miranda, 1991*), but due to deforestation it is now a mosaic with a high percentage of pasture, cropland, fencerows, and isolated trees (K Winker et al., personal observations; *Dirzo & Garcia, 1992*; *Estrada, Coates-Estrada & Merritt, 1997*). *Andrle (1966)* estimated that 50% of the region was forested in 1962. By 1975 *Rappole & Warner (1980)* estimated that a third of the forests still stood. Just 15% of forest remained in 1986 (*Winker, Rappole & Ramos, 1990*; *Dirzo & Garcia, 1992*), and in 1994 only 7%–10% of the region was forested (*Winker, 1997*). Remaining forest occurs primarily in the highlands, and below 500 m forest is scarce (*Rappole, Powell & Sader, 1994*; *Mendoza, Fay & Dirzo, 2005*; Figs. 1 and 2, Fig. S1).

The climate in Los Tuxtlas is warm and wet, with a mean annual temperature of 25 C, and annual precipitation is 4,500–4,900 mm, with a short dry season from March–May (*Soto & Gama, 1997*). Canopy heights in primary forest range from 30 to 35 m (*Ibarra-Manríquez et al., 1997*). Second growth areas generally have variable canopy heights from 3 to 20 m (K Winker et al., personal observations).

In 1967 the Universidad Nacional Autónoma de México established the Estación de Biología Los Tuxtlas, protecting a 640-ha tract of lowland rainforest (*González-Soriano,*

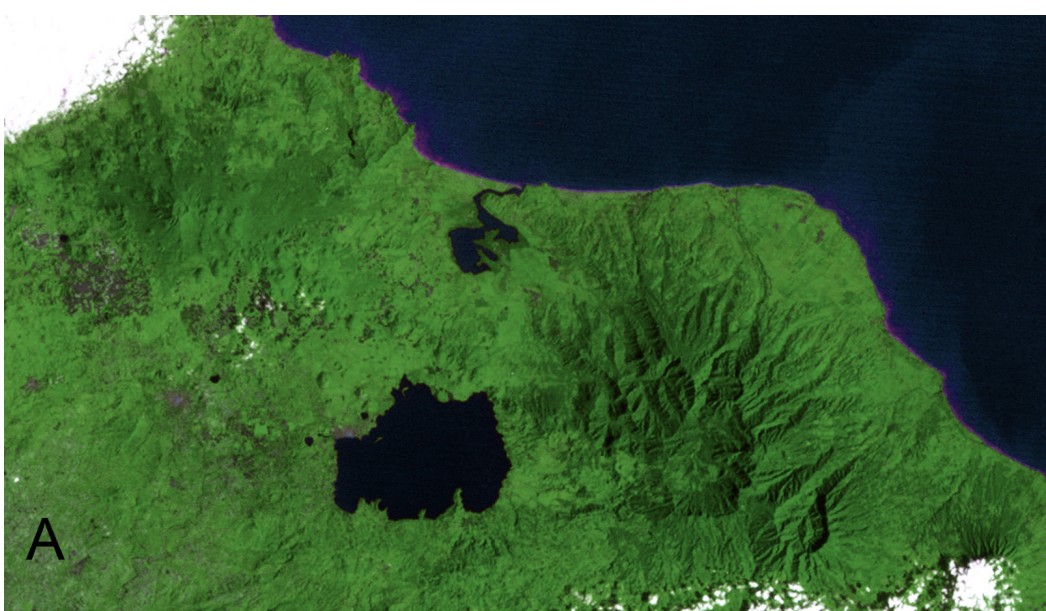

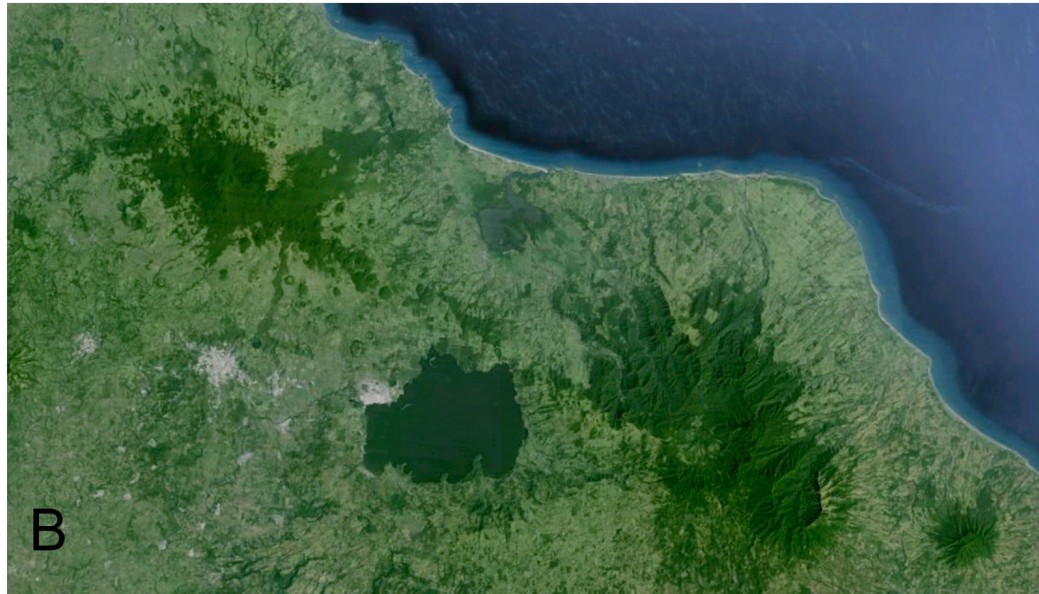

**Figure 1** Comparative views of the Sierra de Los Tuxtlas from an artificially colorized 1979 Landsat image (A) and a 2010/11 Google Earth image (B) showing the extent of deforestation in the region. Remaining forest has become concentrated at higher elevations on the slopes of the region's three volcanoes, San Martín, Santa Marta, and San Martín Pajápan (the forested areas remaining, from left to right).

*Dirzo & Vogt, 1997*). Over the following decades this site became largely isolated from other tracts of forest, although a corridor of forest remains, connecting to the more extensive upland forests on Volcán San Martín (*Dirzo & Garcia, 1992*; Fig. 2). The first intensive sampling of birds in the region began in 1973, and data from that effort are included here (see *Winker, 1997*).

During the non-breeding seasons of 1973–74 and 1974–75 Oehlenschlager, Ramos, Rappole, and Warner conducted the first intensive mist-netting efforts in the area.

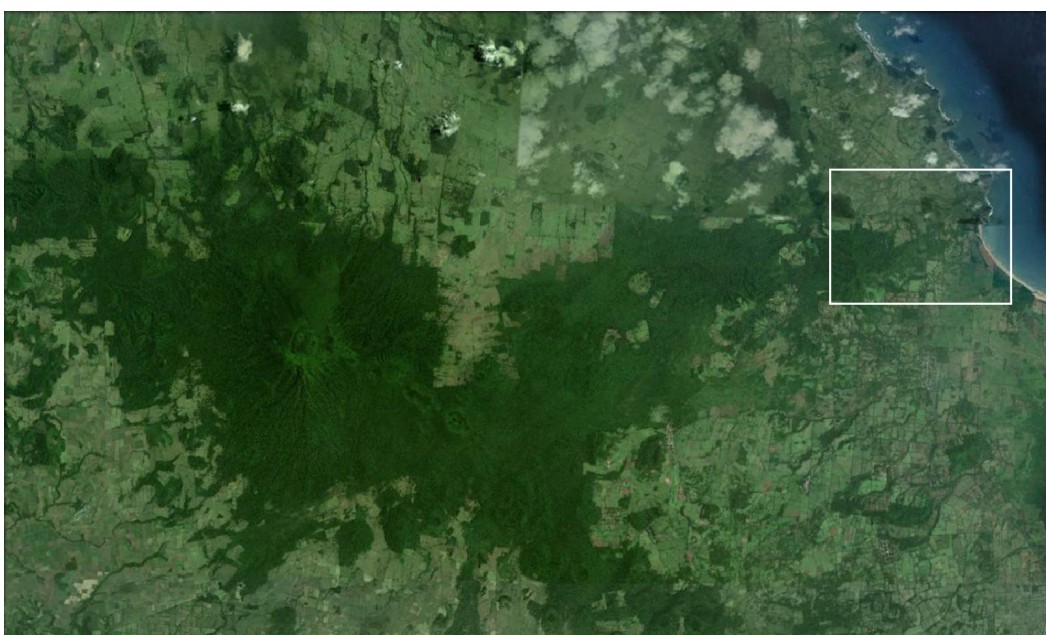

**Figure 2** Satellite view of Volcán San Martín, the northernmost volcano in the Sierra de Los Tuxtlas, showing the distribution of forests (dark areas). The study area is indicated by the white box, which corresponds to the area in Fig. 3 (image from Google Earth, 2010).

**Table 1** Sample effort and periods during eight nonbreeding seasons across three decades in the Sierra de Los Tuxtlas, Veracruz, Mexico.

| Nonbreeding season | Net hours | Sampling period |
|---|---|---|
| (1) 1973–74 | 33,976 | 15 Aug–26 May |
| (2) 1974–75 | 36,512 | 7 Aug–29 May |
| (3) 1986–87 | 4,310 | 17 Nov–16 Jan |
| (4) 1992–93 | 12,605 | 5 Sep–15 Nov |
| (5) 1993–94 | 41,142 | 25 Aug–20 May |
| (6) 1994–95 | 22,509 | 15 Aug–15 Nov |
| (7) 2002–03 | 8,395 | 21 Feb–27 Apr |
| (8) 2003–04 | 2,312 | 5 Apr–29 Apr |

Sites extended through what was then contiguous rainforest from the biological station eastward to the coast (Fig. 3). In 1986, Rappole, Ramos, and Winker operated mist nets at the biological station, and Winker and Escalante continued work there from 1992 to 1994. In 2003–04 as part of a study of migrant birds, Shaw operated mist nets at the same location as Winker and Escalante's work in the 1990s. This study was approved by the University of Alaska Fairbanks IACUC (approval numbers: #00-33 & #04-03). Fieldwork occurred primarily during the non-breeding season. Effort was made to equally sample the available forest types throughout the study period, although, in order to do this, habitat changes precluded using the same sites across all years (see *Winker, 1995*; Fig. 3). Field effort as gauged by net hours also varied among years (Table 1).

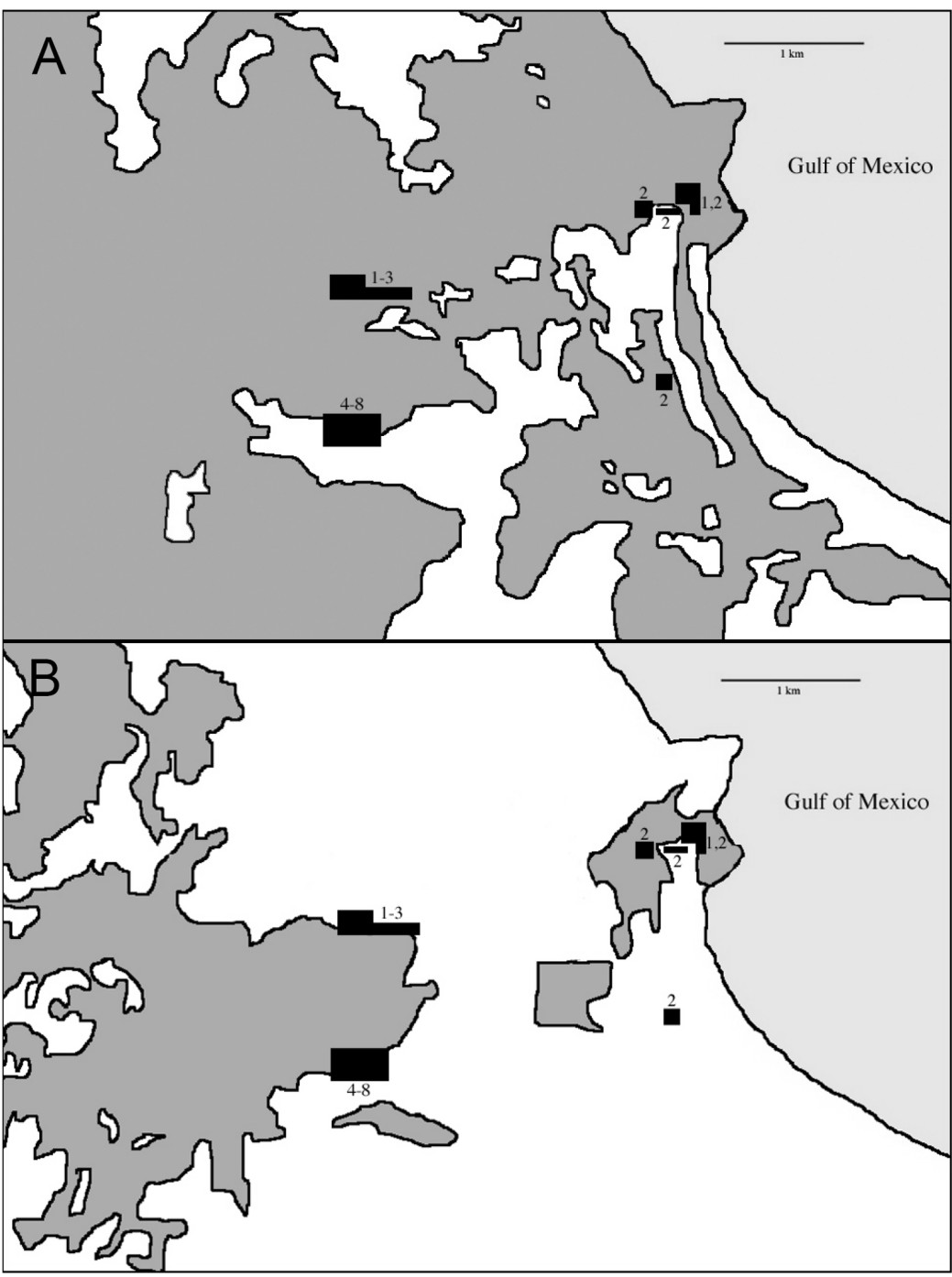

**Figure 3** Maps of the study area in the northern lowlands of the Sierra de Los Tuxtlas (this is the area in the white box in Fig. 2) showing a rough outline of all forests types (dark gray areas) in 1979 (A, from Landsat image), in 2005 (B, from GoogleEarth), and netting sites (black polygons). Numbers indicate field season(s) site was used and correspond to rows in Table 1.

Our earliest sampling occurred over a wider area than later seasons (Fig. 3). During the earliest sampling, large tracts of contiguous forest consisting of various microhabitats dominated the region and were sampled accordingly (Fig. 3). This broader expanse of forest likely provided habitat to more species than the current distribution of forest. This increased detection probabilities for some species such as *Schiffornis turdina*, which was rare even during our earliest sampling. Two general types of forest were present after fragmentation: primary forest and acahual (second growth). Because our sampling was forest-oriented, our efforts tracked the distribution of these habitats. Primary rainforest and second growth habitats were sampled in all efforts. We were unable to separate capture data by site for the early sampling periods; our findings therefore include data from the somewhat larger area from the station east to the coast. Our sampling was also uneven with respect to season, with wet and dry season sampling being unevenly distributed among years; we attempt to account for this, especially in relation to seasonal movements, when considering the results. This sampling heterogeneity leads us to be cautious and conservative in our analyses and interpretations. Importantly, however, the same site (18°34′50″ N, 95°04′20″ W) and net lanes were used in the 1992–2004 efforts (sample periods 4–8 in Table 1).

Only resident species were used in our analyses due to seasonal migration and the high levels of variance in abundance this causes among obligate migrants. Changes in relative abundance were detected by comparing capture rates (birds per 1000 net hours) from each year of sampling. Through visual inspection of data (Appendix S1) we chose species absent in later samples and those with trends of apparently declining or increasing rates of capture for more detailed analyses. Neither gaps nor monotonic changes were necessary for inclusion, just suggestion of a possible trend. We did this instead of applying statistical tests across all 122 species to minimize Type I and Type II errors either by applying a very large number of tests or a conservative correction (e.g., Bonferroni). Presence/absence patterns and observational data (daily checklists in later years) were also considered to provide insight into changes in abundance in low-density species that did not have sufficient samples for statistical testing. Species were considered for examination for presence/absence if they had not been captured since at least 1986–87. Vagrants, defined as those rarely encountered species whose ranges do not normally include the Sierra de Los Tuxtlas, were excluded (*Winker et al., 1992*; *Howell & Webb, 1995*). Only first-time captures (within a season) were used in statistical analyses. Ordinary least squares regression was used to detect changes in abundance for selected species. We looked for newly appearing species using presence/absence netting, observational, and specimen data. Daily checklists were used to augment mist-net data as a check to determine whether absence from the mist-net data was indicative of reality.

Species showing statistically significant declines and those not captured or observed in later sampling periods were categorized by preferred habitat (edge, forest, or semi-open), food preference (fruit/nectar or insects), elevational range, and whether Los Tuxtlas was at the periphery or core of its geographic range (*Howell & Webb, 1995*). These characteristics were used to assess whether certain traits of the species increased their vulnerability to local extirpation.

## RESULTS

During this study we accumulated 165,083 net hours, equivalent to 37.7 net years if netting with a single net occurred twelve hours per day (Table 1). A species accumulation curve for a representative year (1992) with below-average net hours (12,605; mean = 20,220) showed that the avifauna was effectively fully sampled during most field seasons (Fig. S2, though in documenting a species' absence it is the among-season, aggregate sampling that is important). In total, 122 nonmigratory species were captured (Appendix S1).

Seven species showed statistically significant declines during the sampling period: *Phaethornis striigularis, Xenops minutus, Glyphorynchus spirurus,Onychorhynchus coronatus, Myiobius sulphureipygius, Henicorhina leucosticta*, and *Eucometis penicillata* (Table 2). Of these taxa, four were captured throughout the sampling period: *P. striigularis, X. minutus, E. penicillata*, and *H. leucosticta. G. spirurus* was last captured in 1975, *O. coronatus* in 1986, and *M. sulphureipygius* in 1994, the last season of autumn netting. Four other species were captured in substantial numbers during early sampling periods but were not captured in later years: *Lepidocolaptes souleyetii, Ornithion semiflavum, Leptopogon amaurocephalus*, and *Coereba flaveola* (the latter may be an intratropical migrant in this region; *Ramos, 1983*); however, these species failed to show statistically significant declines in linear regression analyses, perhaps due to nonlinear declines. *L. souleyetii* was last captured in 1993–94, and the others were last captured in 1994–95. One species, *Hylomanes momotula,* was captured from 1986–1995 but not in the 1970s or in 2003–04. Though there were no captures in the 1970s, one individual was collected on 17 May 1974 a few km northeast of the station. A similar pattern occurred in *Anabacerthia variegaticeps*, with captures occurring only in the 1990s. Only two species (*Trogon collaris* and *Xiphorhynchus flavigaster*) showed significant increases during the study period.

Presence/absence mist-net capture data for low-density species not captured after 1986–87 could be interpreted as suggesting that an additional 23 taxa were extirpated during the study (Table 3). However, we know from observational data that not all of these species were absent. These taxa included rarely captured species that are too large for effective mist-net capture or that prefer the forest canopy (e.g., *Micrastur ruficollis, Cotinga amabilis*), mixed/open habitat specialists (e.g., *Thraupis abbas* and *T. episcopus*), a small-stream specialist (*Chloroceryle aenea*), and highland species (e.g., *Myadestes unicolor*) that are either not prone to capture in mist nets or at our site. Species such as *Tityra inquisitor*, both *Thraupis* tanagers, and others were known to be present on the site or nearby but were not captured in later sampling periods. Four species of hummingbirds are included in Table 3, but due to inconsistent capture probabilities of low-density hummingbird species and non-definitive observational data with respect to accurate identification, we provide no hypotheses regarding their possible extirpation or persistence at the site; further work focusing on these species is warranted. There were six other species not in Tables 2 or 3 in which mist net data alone might suggest declines or absences (Appendix S1) during the entire study but which were present throughout from observational data; netting is not an effective sampling tool for these taxa because of body size or forest stratum occupied (e.g., *Glaucidium brasilianum, Ciccaba virgata*, and *Celeus*

**Peer**J

**Table 2 Outcomes of regression analyses for 14 species showing changes in abundance (capture rates; captures and rates are given in the Appendix S1) and those not detected in the later sampling periods.** Those $P$-values presented in bold are significant at $\alpha = 0.05$.

| Species | $F$ | $P$ | $R^2$ | Last captured |
|---|---|---|---|---|
| *Phaethornis striigularis*[c] | 6.337 | **0.045** | 0.514 | 2002–03 |
| *Hylomanes momotula*[a] | 0.210 | 0.890 | 0.003 | 1994–95 |
| *Trogon collaris*[b] | 7.041 | **0.038** | 0.540 | n/a |
| *Xiphorhynchus flavigaster*[b] | 6.941 | **0.039** | 0.536 | n/a |
| *Xenops minutus*[c] | 7.578 | **0.033** | 0.558 | 2003–04 |
| *Glyphorynchus spirurus*[c,d] | 7.529 | **0.034** | 0.557 | 1974–75 |
| *Lepidocolaptes souleyetii*[d] | 3.265 | 0.121 | 0.352 | 1992–93 |
| *Ornithion semiflavum*[d] | 0.327 | 0.588 | 0.052 | 1994–95 |
| *Leptopogon amaurocephalus*[d] | 2.814 | 0.144 | 0.319 | 1994–95 |
| *Onychorhynchus coronatus*[c,d] | 6.861 | **0.040** | 0.533 | 1986–87 |
| *Myiobius sulphureipygius*[c,d] | 10.555 | **0.019** | 0.629 | 1994–95 |
| *Henicorhinal eucosticta*[c,d] | 6.740 | **0.041** | 0.529 | 2003–04 |
| *Coereba flaveola*[d] | 2.164 | 0.192 | 0.265 | 1994–95 |
| *Eucometis penicillata*[c] | 18.725 | **0.005** | 0.757 | 2002–03 |

**Notes.**
[a] Species captured 1986–1995. See text.
[b] Species showing an increase in abundance.
[c] Species showing a significant decline.
[d] Species not captured in later sampling periods.

*castaneus*) or because forest understory is not preferred habitat (e.g., *Pitangus sulphuratus, Myiozetetes similis*, and *Volatinia jacarina*; Appendix S1). The first three of these species require more focused study to determine abundances and possible declines.

Four lower-density species have likely been extirpated: *Taraba major, Formicarius analis, Grallaria guatimalensis*, and *Schiffornis turdina* (Table 3). One low-density species that might seem to have been extirpated from our data, *Elaenia flavogaster*, is likely an intratropical migrant here (K Winker et al., personal observations; *Howell & Webb, 1995*; Table 3). Several species were captured only in later sampling periods (Appendix S1) but were observed or collected throughout, suggesting that there were no additions to the biological station's resident avifauna during the study.

Based on all available data during the study (netting and observational data), a minimum of 11 species of birds appear to have been extirpated from the biological station over the past three decades. This translates into an average loss of 3.7 species per decade or a local loss of 2.0% of the entire Los Tuxtlas avifauna (561 spp.; *Schaldach & Escalante, 1997*), 4.1% of the resident avifauna (269 spp.; *Schaldach & Escalante, 1997*), or 9.0% of the resident species captured in our study (122 spp.; Appendix S1). All 16 species showing significant declines or no longer present on the site prefer some degree of forest cover (Table 4). Three species are edge specialists: *O. semiflavum, O. mexicanus*, and *C. flaveola*. Eleven prefer closed canopy forest: *P. striigularis, H. momotula, X. minutus, G. spirurus, F. analis, G. guatimalensis, L. amaurocephalus, M. sulphureipygius, S. turdina,*

**Table 3 Species not captured or observed from 1992–2004, seasons captured (from Appendix S1), presence on the field site in later sampling periods, and comments.**

| Species | Seasons captured | Presence | Comments |
| --- | --- | --- | --- |
| *Micrastur ruficollis* | 1 | Y | observed |
| *Crypturellus boucardi* | 3 | Y | observed |
| *Heliomaster longirostris* | 1 | ? | hummingbird |
| *Florisuga mellivora* | 1 | ? | hummingbird |
| *Chlorostilbon canivetii* | 2 | ? | hummingbird |
| *Hylocharis eliciae* | 1, 2 | ? | hummingbird |
| *Chloroceryle aenea* | 1, 2 | Y | small streams |
| *Dryocopus lineatus* | 2 | Y | observed |
| *Synallaxis erythrothorax* | 2 | Y | observed |
| *Taraba major* | 2 | N | forest understory |
| *Formicarius analis* | 1 | N | forest understory |
| *Grallaria guatimalensis* | 1, 3 | N | forest understory |
| *Tityra inquisitor* | 1 | Y | observed, canopy |
| *Cotinga amabilis* | 1 | ? | canopy |
| *Schiffornis turdina* | 1 | N | forest understory |
| *Polioptila plumbea* | 1 | Y | observed |
| *Myadestes unicolor* | 1 | Y | highlands |
| *Euphonia affinis* | 2 | ? | none |
| *Thraupis abbas* | 1 | Y | observed |
| *Thraupis episcopus* | 2 | Y | observed |
| *Saltator atriceps* | 1, 2 | Y | observed |
| *Molothrus aeneus* | 1 | Y | observed |

*H. leucosticta*, and *E. penicillata*. *T. major* prefers primary forest edge, second growth, and riparian thickets, while *L. souleyetii* prefers semi-open or partly cleared forest.

Eleven of 16, or 68.8%, of the species showing declines or extirpations in this study are insectivores, whereas among all species captured 41% are insectivores. This trend was not significant, however (*G*-test with Williams' correction, $P > 0.1$).

The Sierra de Los Tuxtlas is the northernmost limit of the ranges of 13 of the 16 species showing declines. *G. guatimalensis* and *H. leucosticta* are the only species with a distribution extending substantially to the north and west of the study site. The field site is well within the elevational limits for all 16 species (Table 4).

The two species that significantly increased in abundance over the sample period (Table 4) both occur here at the core of their ranges, elevational distributions, and in their preferred forest habitat. *T. collaris* is a frugivore, and *X. flavigaster* is an insectivore.

## DISCUSSION

Although the absence of a species is not a clear indication of extirpation, our sampling effort, despite its heterogeneity, does suggest that at minimum a species' absence indicates a decline. It is possible that some of the species now apparently gone from the station may persist in other, unsampled fragments. If the data presented here and our interpretations of

**Table 4 Habitat, foraging preference, elevational range, and position within geographical distribution for 18 species of birds at the Estación de Biología Los Tuxtlas (from *Howell & Webb, 1995*).**

| Species | Habitat preference | Foraging guild | Elevational distribution (m) | Geographic distribution |
|---|---|---|---|---|
| *Phaethornis striigularis* | forest | nectarivore | 0–1500 | periphery |
| *Hylomanes momotula* | forest | frugivore | 0–1500 | periphery |
| *Trogon collaris* | forest | frugivore | 0–2400 | core |
| *Xenops minutus* | forest | insectivore | 0–1000 | periphery |
| *Xiphorhynchus flavigaster* | forest | insectivore | 0–1500 | core |
| *Glyphorynchus spirurus*[*] | forest | insectivore | 0–1200 | periphery |
| *Lepidocolaptes souleyetii* | semi-open | insectivore | 0–1500 | periphery |
| *Taraba major*[*] | forest | insectivore | 0–1600 | periphery |
| *Formicarius analis*[*] | forest | insectivore | 0–750 | periphery |
| *Grallaria guatimalensis*[*] | forest | insectivore | 50–3500 | core |
| *Ornithion semiflavum* | edge | insectivore | 0–1500 | periphery |
| *Leptopogon amaurocephalus* | edge | insectivore | 0–1300 | periphery |
| *Onychorhynchus coronatus* | forest | insectivore | 0–1200 | periphery |
| *Myiobius sulphureipygius* | forest | insectivore | 0–1000 | periphery |
| *Schiffornis turdina*[*] | forest | frugivore | 0–750 | periphery |
| *Henicorhina leucosticta* | forest | insectivore | 0–1300 | core |
| *Coereba flaveola* | edge | frugivore | 0–1000 | periphery |
| *Eucometis penicillata* | forest | frugivore | 0–750 | periphery |

**Notes.**

[*] Presence/Absence data suggest species is extirpated.

them are accurate, the extirpation of species from the Estación de Biología Los Tuxtlas has been ongoing since its isolation. Such an "extinction debt" is a recognized component of deforestation, and models of empirical data show that in birds this occurs across decades, but the species affected and the mechanisms of species loss remain poorly understood (*Tilman et al., 1994*; *Ewers & Didham, 2006*; *Robinson & Sherry, 2012*). Since 1973, 16 species susceptible to capture in mist nets have either become locally extirpated or are showing significant declines in abundance. The total number of losses and declines is undoubtedly higher than presented, because species not regularly captured in mist nets, such as large-bodied and canopy species, were not adequately surveyed. Species known to have been extirpated from Los Tuxtlas include *Sarcoramphus papa*, *Harpia harpyja*, and *Ara macao*. *Patten, Gómez de Silva & Smith-Patten (2010)* also documented the extirpation of the latter two in Chiapas, Mexico. Many additional species have also been categorized as endangered or threatened in the Sierra de Los Tuxtlas (see *Winker, 1997*).

Our estimate of the average rate of avian losses from the station of 3.7 species per decade may not be directly comparable to other studies due to differences in habitat and sampling, but it is similar to the rate of loss observed at Barro Colorado Island by *Robinson (1999)* of 3.3 species per decade. Our estimate, however, includes only those taxa captured in mist nets, whereas Robinson's work included all species detected through observation.

Peer J _________________________________________________

Of the eight species with data sufficient for statistical analysis that showed local extirpation, six were lost between 1992 and 2004 (on the same site), suggesting a continuing extirpation of species from the station. *Bierregaard & Lovejoy (1988)* and *Bierregaard & Lovejoy (1989)* found that as surrounding habitat was lost, species richness in remaining fragments increased as individuals displaced from surrounding areas found their way to remaining forest patches. This increased richness was limited by the lifespan of the individual birds (*Bierregaard & Lovejoy, 1988*; *Bierregaard & Lovejoy, 1989*). Unlike these studies, in which forest patches were suddenly and completely isolated, the forest of the Estación de Biología Los Tuxtlas was isolated gradually. Because extirpation seems to be continuing, we expect declines and extirpations to continue for some time at the station, even if no further deforestation occurs in the region (*Willis, 1974*; *Brooks, Pimm & Oyugi, 1999*; *Robinson, 1999*; *Ferraz et al., 2003*).

Mechanisms for tropical bird species losses due to deforestation and fragmentation probably include factors such as greater specialization as compared to temperate birds, reduced dispersal abilities, lower population densities, and patchy distributions (*Robinson et al., 2004*; *Stratford & Robinson, 2005*; *Moore et al., 2008*; *Rompré et al., 2007*). Our assessment of possible causes for the loss of these species reveals no definite patterns, however, other than the predominant requirement of forested habitat. On Barro Colorado Island in Lake Gatún, Panama, maturation of habitat and loss of open areas was responsible for the decline in the island's avifauna (*Willis, 1974*; *Karr, 1982*). This is unlikely to be the case in Los Tuxtlas. Despite major degradation of surrounding forests, the station has remained primary forest with areas of second growth. A loss of sapling and seedling species has been described (*Dirzo & Miranda, 1990*), but the overall structure of the forest appears to have remained fairly stable. *Vetter et al. (2011)* found in a meta-analysis of 30 studies that the effects of fragmentation are not subject to simple generalities, and that they are highly site specific. *Patten & Smith-Patten (2011)* pointed to the need to understand extirpations at local scales because responses can differ from predictions made at larger scales.

Los Tuxtlas is at the northernmost extent of the ranges of 13 of the 16 species we found to be declining or extirpated (Tables 3 and 4). Evidence is mixed as to whether populations at the periphery of a species' range are more vulnerable to extirpation (*Terborgh & Winter, 1980*; *Kattan, Alvarez-López & Giraldo, 1994*; *Johnson, 1998*). Los Tuxtlas is at the edge of all species' geographic ranges endemic to Neotropical rainforest, so it is not clear why this subset might be more subject to this phenomenon. The elevational distribution of each of these species encompasses sea level to 750 m or more (*Howell & Webb, 1995*), and we consider this factor unlikely to be responsible for the vulnerability of these particular taxa.

Although insectivores showed a trend toward being disproportionately affected in our study, it was not significant. Elsewhere insectivores have been shown to be particularly vulnerable to severe habitat change (e.g., *Kattan, Alvarez-López & Giraldo, 1994*; *Canaday, 1996*; *Johnson & Winker, 2010*; *Vetter et al., 2011*). Additionally, deforestation can negatively impact species found in multi-species foraging flocks (*Van Houtan et al., 2006*), which are important to many birds of tropical rainforest communities (*Willis, 1966*; *Morton, 1973*;

*Buskirk, 1976*; *Rappole et al., 1983*). *Rappole & Morton (1985)* noted that *X. minutus*, one of the species showing a significant decline in our study (Table 2) was a regular member of mixed flocks in the Sierra de Los Tuxtlas.

We considered large-scale range shifts, perhaps from climate change, as a possible cause for species loss, but this seems unlikely. At least some of the species lost in our study appear to have persisted in the southern portion of Los Tuxtlas near Volcán Santa Marta at least into the mid-1990s (K Winker, personal observations). If range shifts were the cause, species would likely have disappeared region-wide and we would not expect only forest-related species to be affected. Habitat loss and degradation seem to be the best explanations for the losses observed, but exactly how these changes affected each species remains unknown.

Another possible influence on mist-net captures, particularly in the most recent, late winter/spring sampling periods, would be seasonal intra-tropical and elevational movements in some of the study species (*Ramos, 1983*; *Ramos, 1988*). There is evidence that *C. flaveola* and *E. flavogaster* move seasonally within the tropics, seemingly to breed in Los Tuxtlas then departing (M Ramos & J Rappole, personal observations). *Vega Rivera (1982)* found probable elevational movements in *M. sulphureipygius*. The extirpations of seven of the 16 species are particularly notable. *C. flaveola* is a widely distributed species known to thrive in manipulated habitats such as gardens and forest edges and is a generalist frugivore and nectarivore (K Winker et al., personal observations; *Howell & Webb, 1995*). This is not a species we would expect to decline due to forest fragmentation; both its habitat and food preferences are well suited to survival in a mosaic landscape, and it is known to persist in a fragmented landscape elsewhere in northern Middle America (*Johnson & Winker, 2010*). Intratropical migrations of *C. flaveola* may partially explain the changing capture rates in this species (M Ramos & J Rappole, personal observations). *O. semiflavum* and *L. amaurocephalus* are both edge specialists; thus, limited fragmentation, creating an increase in edges, might *a priori* seem to benefit these species. Though the habitat protected by the station has remained relatively static, the intensity of lowland deforestation in Los Tuxtlas as a whole (Fig. 1) may be too extensive even for these edge specialists. *L. souleyetii* prefers open forest and partially cleared areas (*Howell & Webb, 1995*). The habitat surrounding the station during the 1980s and 1990s was dominated by pasture scattered with isolated trees. In our later field seasons there was a noticeable decline in the number of isolated trees and fences constructed of living trees (K Winker, personal observations). This loss may account for the extirpation of *L. souleyetii*. *G. spirurus* apparently disappeared from the station between the 1970s and 1986, the first of the documented extirpations. The majority of deforestation across the region took place during this period. This previously abundant species disappeared from our data in just over a decade. Interestingly, on the slopes of neighboring Volcán Santa Marta the species was present at least into the 1990s and probably still persists there (K Winker, personal observations). Also, *Estrada, Coates-Estrada & Merritt (1997)* had observational data of the species' presence in the station area in 1990–1992, indicating at least a decline if not extirpation (Table 2). In Brazil, *G. spirurus* persisted in experimentally isolated

fragments well after isolation (*Stouffer & Bierregaard, 1995*), and the species persists in highly fragmented forest in southern Belize (*Johnson & Winker, 2010*). *H. momotula* was collected but not netted in 1974, was captured in substantial numbers during 1986 and 1992–94, but was absent in the last two seasons of sampling. This pattern is mysterious. This species has an elevational range extending to 1500 m and may persist in the forests of the upper slopes of Volcán San Martín. If so, we speculate that the station may serve as a sink for this species, where habitat is insufficient for a self-sustaining population but may occasionally be colonized by dispersing individuals (see also *Winker et al., 1997*). Continued sampling may provide more insight into its abundance patterns. It illustrates the need for improved understanding of species-specific dispersal behavior within and among forest fragments (e.g., *Van Houtan et al., 2007*; *Moore et al., 2008*; *Ibarra-Macias, Robinson & Gaines, 2011*), which may be an important driver for patterns such as those we observed.

Two other studies provide comparative value to our results. The four species we consider likely extirpated (*Taraba major*, *Formicarius analis*, *Grallaria guatimalensis*, and *Schiffornis turdina*) were not detected in the much broader census surveys of *Estrada, Coates-Estrada & Merritt (1997)* in 1990–1992. *Patten, Gómez de Silva & Smith-Patten (2010)* conducted the geographically closest long-term study to ours in their analysis of avian declines at Palenque, Chiapas, Mexico. Their results showed only three species that overlapped our results. They found *Eucometis penicillata* extirpated (to our decline) and two others that declined as our populations did (*Xenops minutus* and *Leptopogon amaurocephalus*). Indeed, the species-level heterogeneity between our studies is noteworthy. A key similarity between our studies, however, is the importance of forest in explaining declines and extirpations (*Patten & Smith-Patten, 2011*).

Our analyses suggest that the Estación de Biología Tropical Los Tuxtlas is too small to maintain its full, historic complement of bird species. If deforestation accelerated region-wide, eliminating other forest refugia, the station alone (640 ha) would be unable to maintain the historical avian diversity of the region or to provide source populations for restored forest habitats for many of its present bird species. Given the scale of deforestation in the region, it is surprising that there are not more species showing declines. Indeed, we may consider it good news that important forest seed dispersers such as *Habia* tanagers (*Puebla & Winker, 2004*) did not show significant declines. The overall size of the remaining forests, particularly in the highlands, may be ameliorating the effects of lowland deforestation. However, increasing or continued isolation of the station will probably limit recolonization from elsewhere, and species losses will likely continue.

In our study, although several species seemed to quickly succumb to local and regional deforestation, others showed delayed declines and extirpations, a phenomenon also known to happen at larger scales (*Tilman et al., 1994*; *Pimm et al., 2006*). Moreover, the effects of deforestation were remarkably heterogeneous among forest-related species, with no single clear pattern of why some species experienced declines or extirpation. Our long-term data suggest that predicting which species will be most affected by deforestation in the northern Neotropics, and thus effectively working to ameliorate the effects of forest loss, will be

particularly challenging. Nevertheless, as similar long-term datasets accrue, subtle patterns may reveal how species-specific responses reflect underlying commonalities that can be exploited for effective management and conservation.

## ACKNOWLEDGEMENTS

We thank the many field assistants who have helped us over the years and those who have provided and helped obtain permits to conduct this research. We also thank Len Kamerling for assistance with images. R Barry, A Powell, S Pimm, M Patten, and three anonymous reviewers provided excellent advice and comments.

### Funding

The University of Alaska Museum and the Friends of Ornithology provided financial support for this study in 2002 and 2004. Earlier seasons were supported by the Welder Wildlife Foundation, the World Wildlife Fund (US), the Smithsonian Institution, Friends of the National Zoo, the Wildlife Conservation Society, and the Chicago Zoological Society. This study is also part of project No. 109298 funded by FOMIX (CONACyT-VERACRUZ). The funders had no role in study design, data collection and analysis, decision to publish, or preparation of the manuscript.

### Grant Disclosures

The following grant information was disclosed by the authors:
University of Alaska Museum and the Friends of Ornithology.
Welder Wildlife Foundation, World Wildlife Fund (US), Smithsonian Institution, Friends of the National Zoo, Wildlife Conservation Society, and Chicago Zoological Society.
FOMIX (CONACyT-VERACRUZ).

### Competing Interests

The authors declare that they have no competing interests.

### Author Contributions

- David W. Shaw and Richard J. Oehlenschlager performed the experiments, analyzed the data, wrote the paper.
- Patricia Escalante, John H. Rappole and Kevin Winker conceived and designed the experiments, performed the experiments, analyzed the data, wrote the paper.

### Animal Ethics

The following information was supplied relating to ethical approvals (i.e., approving body and any reference numbers):

This study was approved by the University of Alaska Fairbanks IACUC (approval numbers: #00-33 & #04-03).

## Supplemental Information

Supplemental information for this article can be found online at http://dx.doi.org/10.7717/peerj.179.

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
