# Peer review of "Decadal changes and delayed avian species losses due to deforestation in the northern Neotropics"

_PeerJ, doi:10.7717/peerj.179_

## Round 0.1 · original submission · Minor Revisions

The reviewers suggest a few minor corrections clarifying certain elements of the paper and related to the figures which I'm confident came be dealt with swiftly.

·

Basic reporting

The authors analyzed mist net captures gathered at various times between 1973 and 2004 to deduce population changes in bird species in the Sierra de los Tuxtlas in southeastern Mexico. On the basis of these data, coupled with presence/absence data from direct detection in the field (i.e., records of species not mist netted but detected during the netting effort), they conclude that a dozen species have declined or been extirpated from the Tuxtlas.

My comments are relatively minor given that this manuscript is straightforward in its goals and results. For example, I could see merit, somewhere in the second paragraph of the Introduction, in bringing in ecological notions of resistence and resilience, notions that may help to guide predictions. Otherwise, my thoughts center on what I see a need to be clearer with some of the methods, chiefly to aid repeatability.

Experimental design

Key among my concerns is the specific meaning of “apparently declining or increasing rates” (line 174). Because not all species were tested, it is imperative that anyone who wishes to repeat the results understands what “apparent declines” are. Was this, for instance, a monotonic decrease in capture frequency? Or was there a need for gaps in the capture record? And what is meant by use of only “first-time captures” (l. 183): was this within a season (in which case I understand the policy) or over the life of the study (in which case I do not)? If the latter, a bird banded in, say, 1995 would not be counted if it was recaptured in 2002, even though its very presence indicates persistence.

If I had any criticism that tended toward “major” it concerns the seasonality of the effort. A perusal of Table 1 shows that the mist net data were gathered in both the wet and dry seasons in the 1970s, chiefly during the wet season in the 1980s and in 1992-1993, in both in 1993–1994, in the wet season in 1994–1995, and in the dry season in the 2000s. There is an enormous amount of intra-tropical movement of birds in Middle America with respect to timing of wet and dry seasons. Some species barely move, whereas others (e.g., many hummingbirds, Cotinga amabilis, Hylomanes momotula, etc.) may be largely absent from a site during the wet season but conspicuous during the dry, or vice versa. The authors acknowledge this effect in some cases, as when they puzzle over the apparent decline of Elaenia flavogaster (l. 254). I would be hard pressed to predict extirpation of this species given that it readily occupies human settlements, so seasonal shifts are a plausible reason for the apparent trend. To what extent might such shifts—about which we know little more many species—account for at least some of the patterns reported in Table 2?

Validity of the findings

no comments (but see those seasonality, above)

Additional comments

l. 32: I think of the term “extinction” as a absolute, just like “pregnant” and “unique.” Hence, I would prefer use of the term “extinction” to be restricted to bona fide extinction (i.e., the utter disappearance of a species). To my way of thinking, “extirpation” is a term that means “local extinction” and so can be used in its place. But this may be little more than a pet peeve or a bucking against an unchangeable trend, so the authors are free to ignore this comment if they feel strongly otherwise.

l. 64: change “how” to “how or if”

l. 158: spell our Schiffornis (and update the specific name to conform to the latest AOU supplement)

l. 183: Does “simple linear regression” mean “ordinary least squares regression”? If so, the latter is the standard term.

l. 199+: Rarefaction would have been superior to simple accumulation curves.

l. 206 (and throughout): correct spelling of striigularis (delete the first “l”)

l. 269: I don’t mean to nitpick, and perhaps it is only a difference in judgement, but I do not think of Taraba major as a forest bird. I agree that it occurs in closed-canopy forests, but I find it just as often at edges or in second growth, including, at least on occasion, in isolated patches of vegetation in fields.

·

Basic reporting

One of my few concerns about this paper is the figures, particularly the grey scale satellite images. Since, this is an online journal, having those in colour would add nothing to the cost of production. Doing so, would greatly enhance the extraordinary story that this paper tells. As I explain below, habitat fragmentation is the major driver of species extinctions and the authors have one of a handful of very well documented cases. So, please, more images and, if possible of intermediate years. Second, if the images could be larger ones, then good. Third, and vitally important, the paper needs better connections to the summary maps of where the study sites are found. It's not clear to me where figure 2 is in relationship to figure 3 — I think it's off to the east. Then, I don't know where figure 3 is in relationship to figure 2 either.

Experimental design

Habitat loss and fragmentation is the major driver of species extinction. But it's a large-scale, long-term process so the numbers of good studies one can count on the fingers of one hand. This is one of them! It's an extraordinary experiment; albeit of a serendipitous kind. The field work is unusually extensive. It involves a massive amount of field work, conducted over decades.

Validity of the findings

This is very important paper, with implications for the species in forest fragments worldwide. The authors provide detailed, but approrpriate, comparisons with the few other similar studies done elsewhere.

Additional comments

Excellent job, but telling the history of land use change in this area needs the very best you can do with satellite imagery and GIS.

---

## Round 0.2 · accepted · Accept

Thanks for swiftly implementing the corrections. I did note that you spelt extirpation wrong in the first sentence but the rest of the text is fine. This can be corrected at the proof stage but it is worth a good look over the proof when it arrives.